# Physiologic and Transcriptomic Effects Triggered by Overexpression of Wild Type and Mutant DNA Topoisomerase I in *Streptococcus pneumoniae*

**DOI:** 10.3390/ijms242115800

**Published:** 2023-10-31

**Authors:** Miriam García-López, Pablo Hernández, Diego Megias, María-José Ferrándiz, Adela G. de la Campa

**Affiliations:** 1Unidad de Genética Bacteriana, Centro Nacional de Microbiología, Instituto de Salud Carlos III, Majadahonda, 28220 Madrid, Spain; miriam.garcialopez@outlook.com; 2Centro de Investigaciones Biológicas Margarita Salas, Consejo Superior de Investigaciones Científicas, 28040 Madrid, Spain; p.hernandez@cib.csic.es; 3Unidad de Microscopía Confocal, Instituto de Salud Carlos III, Majadahonda, 28220 Madrid, Spain; diego-megias@isciii.es; 4Presidencia, Consejo Superior de Investigaciones Científicas, 28006 Madrid, Spain

**Keywords:** DNA supercoiling, DNA topoisomerase I, seconeolitsine, regulation of supercoiling, transcriptional domains

## Abstract

Topoisomerase I (TopoI) in *Streptococcus pneumoniae*, encoded by *topA*, is a suitable target for drug development. Seconeolitsine (SCN) is a new antibiotic that specifically blocks this enzyme. We obtained the *topARA* mutant, which encodes an enzyme less active than the wild type (*topAWT*) and more resistant to SCN inhibition. Likely due to the essentiality of TopoI, we were unable to replace the *topAWT* allele by the mutant *topARA* version. We compared the in vivo activity of TopoIRA and TopoIWT using regulated overexpression strains, whose genes were either under the control of a moderately (P_Zn_) or a highly active promoter (P_Mal_). Overproduction of TopoIRA impaired growth, increased SCN resistance and, in the presence of the gyrase inhibitor novobiocin (NOV), caused lower relaxation than TopoIWT. Differential transcriptomes were observed when the *topAWT* and *topARA* expression levels were increased about 5-fold. However, higher increases (10–15 times), produced a similar transcriptome, affecting about 52% of the genome, and correlating with a high DNA relaxation level with most responsive genes locating in topological domains. These results confirmed that TopoI is indeed the target of SCN in *S. pneumoniae* and show the important role of TopoI in global transcription, supporting its suitability as an antibiotic target.

## 1. Introduction

*Streptococcus pneumoniae* is a major human pathogen. It is the most important etiological agent of community-acquired pneumonia, and main cause of meningitis, bacteremia and otitis media in children, being responsible annually for the death of one million children worldwide [1]. Resistance of this bacterium to beta-lactams and macrolides [2] has spread, which has resulted in recommending fluoroquinolones, which target type II topoisomerases, for treatment [3]. These enzymes, which cleave both strands of the DNA transiently, are topoisomerase IV (TopoIV) and gyrase. Although no worrying levels of resistance to fluoroquinolones have yet detected, they could increase in tandem with an increase in their use, either due to alterations in their targets [4,5,6] or to the action of active efflux [7,8]. In this situation, finding new drug targets against *S. pneumoniae*, and other pathogenic bacteria, is an urgent clinical need. Topoisomerase I (TopoI) is the only pneumococcal type I topoisomerase and cleaves only one strand of the DNA. TopoI and gyrase are the main enzymes that maintain DNA topology in this bacterium [9,10]. TopoI is a suitable new antibacterial target [11] and SCN is a catalytic inhibitor of the TopoI cleavage reaction of *S. pneumoniae* [12,13] and *Mycobacterium tuberculosis* [14]. SCN shows higher bactericidal activity than fluoroquinolones, against both planktonic bacteria and biofilms [13], and is effective against pneumococcal isolates resistant to fluoroquinolones in a murine sepsis model [15].

The regulation of the transcription of topoisomerase genes (*gyrA* and *gyrB* for gyrase; *parE* and *parC* for TopoIV, and *topA* for TopoI) controls homeostasis of supercoiling (Sc) in bacteria. In *Escherichia coli*, DNA relaxation causes a decrease in the transcription of the TopoI gene [16], and an increase in transcription of gyrase genes [17,18]. In addition, several nucleoid-associated proteins, through alteration of Sc [19,20], also affect the expression of *E. coli* topoisomerases. In *S. pneumoniae*, among the three nucleoid-associated proteins identified so far (HU [21], SMC [22], and StaR [23]), HU and StaR are involved in Sc regulation. In addition, in this bacterium, global transcriptomic responses have been detected under DNA relaxation [9] or hypernegative Sc [24], revealing Sc domains. Genes of these domains show a coordinated transcriptional response. DNA relaxation modulates the transcription of 37% of the genome, with a majority (>68%) of responsive genes clustered in 15 up-regulated (UP) or down-regulated (DOWN) domains [9]. On the other hand, hypernegative Sc modulates the transcription of 10% of the genome, with 25% of responsive genes grouped into 12 Sc domains [24]. Domains are enriched in specific functions [25], and their location is conserved in the *Streptococcus* genus [26], suggesting a topology-driven evolution. 

Location of topoisomerase genes in Sc domains determines its transcriptional regulation [26]: *topA* is located in a DOWN-domain; *gyrB* is located in an UP-domain. In this way, Sc regulates transcription, and transcription is at the same time, a major contributor to the level of Sc. The twin supercoiled-domain model proposes that domains of negative and positive Sc are transiently generated behind and ahead of the moving RNA polymerase, respectively [27]. In vitro studies support both this model and a role for TopoI in removing R-loops, which would otherwise interfere with transcription elongation [28,29,30]. In addition, physical interaction of TopoI and RNA polymerase has been detected in vitro for *E. coli* [31] and *S. pneumoniae* [32]. Furthermore, ChIP-Seq experiments have shown in vivo co-localization of RNAP, TopoI and gyrase on the active transcriptional units of *M. tuberculosis* [33] and genome-wide proximity between TopoI and RNA polymerase in *S. pneumoniae*, supporting the interplay between transcription and Sc [32].

In the present study, to ascertain that TopoI is the target of SCN by genetic methods, we have obtained a mutant (*topARA*) of the TopoI coding gene. We analyzed the in vitro activity of TopoIRA and its resistance to SCN. We also compared the in vivo activity of TopoIRA and TopoIWT using regulated overexpression strains, by analyzing Sc and susceptibility to NOV and SCN. Giving the role of TopoI in transcription, we have also studied the transcriptome under over-expression conditions of the *topARA* and wild-type alleles and analyze the Sc domains under NOV treatment. 

## 2. Results

### 2.1. A TopoI Mutant Enzyme Shows Decreased Activity and Increased SCN Resistance In Vitro

A BLAST was made with R6 TopoI against the *S. pneumoniae* sequences available in the NCBI data bank (accessed on 18 March 2022). Among the first 200 sequences producing significant alignments, 131, which had complete TopA and GyrA sequences, were selected for the analysis. The comparison of amino acid sequences of TopoI proteins available in the NCBI data bank revealed that, among 131 sequences, 127 showed variation with respect to the sequence of the R6 strain (Figure 1). Amino acid variations were from 1- to 5-residues of the 701-residue TopoI protein, representing a frequency lower than 0.8%, in agreement with the essential nature of the protein. Likewise, the variation in GyrA of the same strains was in the same range (lower than 0.5%), varying between 1- and 4- changes in the 823-residue GyrA protein.

These changes were unevenly distributed in the TopoI sequence, where domain II showed the highest variation (184 changes in 9 out of 139 residues), perhaps because it does not form part of the gate strand binding site. The remaining domains, which are part of the DNA-binding site, showed lower variations: 2 changes in 2 out of 129 residues (domain I); 37 changes in 6 out of 129 residues (domain III), and 4 changes in 2 out of 106 residues (domain IV) (Figure 1 and Figure 2a).

We unsuccessfully attempted to obtain SCN-resistant mutants using a PCR/transformation approach that previously allowed us to obtain mutants resistant to several chemically unrelated antibiotics [34]. We chose to perform site-directed mutagenesis of *topA* cloned into pQE1 plasmid in *E. coli* followed by protein purification and an activity assay in the presence of SCN. Mutagenesis was directed to obtain four specific substitutions of three key TopoI residues involved in its interaction with SCN: R102A (domain I), D543A (domain IV), E546A, and E546K (domain IV) [12] (Figure 2b,c). In the structural model of the pneumococcal TopoI based on the structure of the *E. coli* enzyme, these three residues are located in the DNA-binding site. However, of the four substitutions, we were only able to obtain the R102A TopoI mutant. The rest of mutant genes contained internal stop codons. The *topARA* and *topAWT* alleles, carrying a 6×—His tail, were overexpressed and TopoIRA and TopoIWT were purified by affinity chromatography (Appendix A). Their enzymatic activities were compared using relaxation assays of pBR322. The amount of TopoIWT enzyme yielding 50% activity was 54.4 ± 22.8 ng (Figure 3a,c), while it was of 1084 ± 240 ng for the TopoIRA enzyme (Figure 3b,d). This means that the activity of TopoIRA is 20-fold lower than that of TopoIWT. 

Inhibition experiments were performed using purified enzymes at 50% activity (Figure 4). These experiments showed that SCN at a concentration of 6.8 ± 2.0 µM reduced the activity of the TopoIWT enzyme to 50%, while a concentration of 13.4 ± 5.2 µM of SCN was necessary for the same reduction of the TopoIRA enzyme activity, indicating that TopoIRA is 2-times more resistant to SCN inhibition. 

### 2.2. Role of TopoIR102A in Cell Viability and NOV Resistance In Vivo

We were unable to replace the *topAWT* allele with *topARA* in the chromosome, likely due to the essential role of Topo I in cell viability. To ascertain the effect of *topARA* on viability, strains R6P_Zn_*topAWT* and R6P_Zn_*topARA* were constructed, which carried the wild-type *topA* gene and an ectopic copy of *topAWT* or *topARA* under the control of P_Zn_ promoter (Figure 5a). Growth of these strains in the absence or presence of ZnSO_4_ with or without subinhibitory NOV concentrations (0.5 × MIC) was studied. In the absence of ZnSO_4_, i.e. under conditions in which both strains express *topA* from the chromosomal copy, the strains carrying *topAWT* and *topARA* showed, as expected, equivalent growth rates (29.9 min ± 0.7 and 31.5 min ± 2.5, respectively, Figure 5b). However, under conditions of TopoI induction, the strain carrying *topARA* showed a higher duplication time (37.5 min ± 2.2) than that carrying *topAWT* (32.8 min ± 0.4), *p* = 0.022. Under these induction conditions, the amounts of TopoIWT and TopoIRA relative to RpoB detected by Western blot were equivalent (2.0 ± 0.5 and 2.4 ± 0.7, Figure 5c). These results support that the differences in growth between the strains overproducing TopoIWT and TopoIRA, could be attributed to the intrinsic activity of the enzymes, confirming the lower specific activity of TopoIRA in vivo. This effect was confirmed under NOV treatment, which induced down regulation of *topA* from its promoter. Then, under NOV treatment and in the presence of ZnSO_4_, the expression of *topAWT* and *topARA* alleles came mainly from P_Zn_, as observed by Western blot (Figure 5b). In these conditions, strain R6P_Zn_*topARA* grew with a higher duplication time than that of R6P_Zn_*topAWT* (64.7 ± 3.6 versus 53.7 ± 2.1, *p* = 0.01), revealing a greater sensitivity to NOV.

To analyze the in vivo activities of TopoIWT and TopoIRA, we estimate Sc densities (σ) by analyzing pLS1 topoisomers (Figure 5d). This estimation of σ values in the plasmid correlates with nucleoid compaction levels estimated by super-resolution confocal microscopy [10]. The analysis of σ values showed a higher activity for TopoIWT, reflected in the higher relaxation (σ = −0.047 ± 0.002) in the presence of NOV when TopoIWT is expressed compared to TopoIRA (σ = −0.051 ± 0.001), *p* = 0.038. 

### 2.3. TopoIR102A Showed SCN Resistance In Vivo

In order to study the effect of over-expressing TopoIRA and TopoIWT enzymes, their corresponding alleles fused to 6×—His tags were cloned into pLS1ROM, leading to pLS1ROM*topAWT* and pLS1ROM*topARA*. These recombinant plasmids carried *topA* under the control of P_Mal_ promoter, which is partially repressed by 0.8% sucrose (S) and moderately induced by 0.4% sucrose + 0.4% maltose (SM). These plasmids were introduced into strain Δ*topA*P_Zn_*topA*, which is as R6P_Zn_*topA* (Figure 5a) but has a *topA* deletion in its native chromosomal location (Figure 6a). To compare the effect of the expression of *topAWT* or *topARA* on growth, cultures grown in the absence of ZnSO_4_ and in the presence of S to exponential phase (OD_620nm_ = 0.4), were diluted 1000-fold in media without ZnSO_4_ containing either S or SM. Both strains grew with equivalent rates in both media. As expected, the addition of the TopoI inhibitor SCN at 0.5 × MIC or 1 × MIC caused growth inhibition, which was significantly higher in SM medium, when the TopoI enzymes were overexpressed. This result indicates that overproduction of either TopoIWT or TopoIRA enzymes is deleterious. However, the strain expressing *topARA* showed lower growth inhibition, having a lower doubling time (131.7 ± 3.0 min) than the strain carrying *topAWT* (173.2 min ± 21.7), *p* = 0.03 (Figure 6b). This result suggests that the TopoIRA enzyme is more resistant to SCN in vivo. Under these experimental conditions in the absence of ZnSO_4_, the only TopoI enzyme synthesized came from the recombinant plasmids, which showed some escape in S medium (Figure 6c). This is supported by the slightly larger size of the recombinant TopoI due to the 6×—His tail relative to the TopoI enzyme expressed from the chromosome in strain R6 pLS1ROM (Figure 6c, left panel). Quantification of Western blots confirmed the overproduction of the TopoI enzymes cloned into pLS1ROM. Expression values in SM inducing medium compared to the values in S medium were of 9.7- and 10.6-fold for TopoIWT and TopoIRA, respectively (Figure 6c, right panel). Compared to the values obtained with the R6pLS1ROM strain, the net increase in TopoI enzymes over the wild-type situation was of 8.8- (TopoIWT) and 9.1-fold (TopoIRA). 

### 2.4. Transcriptomic Effects Triggered by Overproduction of TopoIWT and TopoIRA

We used RNA-Seq to compare the transcriptome of strain Δ*topA*P_Zn_*topA*, carrying pLS1ROM*topAWT* or pLS1ROM*topARA*, under induction conditions (SM medium) and compared it to their transcriptomes in the absence of induction (S medium). As expected, the expression of *topA* increased in both strains in SM medium, being of about 5-fold for *topAWT* and of 6-fold for *topARA* (Figure 7c). This is consistent with the changes in the protein amount detected by Western blot (Figure 6c, about 9-fold for both enzymes). At the same time, Sc density decreased with the overexpression of TopoIWT or TopoIRA, which changed from −0.064 to −0.054 for TopoIWT (*p* = 0.0002) and from −0.061 to −0.054 (*p* = 0.007) for TopoIRA (Figure 7a). No changes in Sc were observed in strain R6 carrying the pLS1ROM vector (−0.060 versus −0.059). That is, the increased amount of TopoI by induction of its expression was accompanied by the reduction of the values of σ. These differences were significant, compared to the values obtained with the pLS1ROM vector (*p* = 0.0003, for pLS1ROM*topAWT*, and *p* = 0.006, for pLS1ROM*topARA*).

The described increases of TopAWT or TopARA enzymes triggered differential transcriptomic responses (Figure 7). A total of 144 (47 up-regulated, 97 down-regulated) differentially expressed genes (DEGs) were detected in the strain overproducing TopoIWT, while a higher number, 236 (96 up-regulated, 140 down-regulated), were detected in the TopoIRA overproducing strain (Figure 7b), being 109 DEGs common to both strains. The strain overproducing TopoIRA gave rise to a higher number of exclusive DEGs (127) than the one overproducing TopoIWT (35). Genes involved in DNA metabolism were more represented among DEGs under TopoIRA overexpression than under TopoIWT overexpression (9 in RA versus 1 in WT, *p* = 0.001). These included down-regulated genes involved in restriction-modification systems (operons coding *spnIRSM* and *dpnCD*), replication (*recX*), and chromosome segregation (*spoOJ*). It also included the recG gene involved in recombination, which was up-regulated. There was also a difference among genes coding hypothetical proteins (103 in TopoIRA versus 53 in TopoIWT, *p* < 0.001).

Besides *topA*, changes in the expression levels of genes coding DNA topoisomerases were not observed (Figure 7c). Neither were significant alterations observed in the distribution of DEGs in the genome between the two strains (Appendix A). 

### 2.5. Transcriptomic Effects Triggered by Overproduction of TopoIWT and TopoIRA under Relaxation Induced by NOV

Relaxation of Sc by inhibition of DNA gyrase with NOV produces changes in the *S. pneumoniae* transcriptome that reveal the existence of domains whose genes are regulated by Sc [9]. To know the role of TopoI in this response to Sc, we analyzed the DEGs produced by NOV under conditions of non-induction (S medium) or induction (SM medium) of *topoIWT* or *topoIRA* (Figure 8). When the cultures overproducing TopoIWT and TopoIRA were treated with NOV, an extensive transcriptomic response was observed. The number of DEGs produced by NOV in S medium was similar in cells producing TopoIWT (874, ~43% transcriptome) or TopoIRA (866, ~42% transcriptome) and most of these genes were common (772) (Figure 8a). The total number of DGEs produced by NOV under TopoI overproduction conditions was higher than in S medium: 1134 for TopoIWT (~55%) and 1074 for TopoIRA (~53%). On the other hand, the relative numbers of DEGs were similar between the two strains and, as in S medium, most of the DEGs were common (965) (Figure 8b). No significant alterations in the genome distribution of DEGs were observed between the two strains (Appendix A).

We were unable to perform consistent 2-D agarose gel electrophoresis of pLS1ROM*topA*WT and pLS1ROM*topA*RA topoisomers in the presence of NOV. This could be due to the instability of the plasmids under these conditions. Alternatively, we estimated Sc levels using our super-resolution confocal microscopy method, recently described [10]. The estimated σ values for the samples were extrapolated from known σ values of control samples with different degrees of DNA relaxation (Figure 9a). As expected, NOV treatment produced a reduction in Sc, compared to the Sc observed in the absence of antibiotics (Figure 9a). This decrease in Sc was higher when TopoI was overproduced in SM medium. It varied from −0.054 to −0.010 for the strain overexpressing *topAWT* and from −0.029 to −0.006 for the strain overexpressing *topARA (*Figure 9b).

We have previously identified topological domains, which are constituted by genes showing a coordinated transcriptional response under topological stress (DNA relaxation with NOV treatment) [24]. Domains were identified considering they should contain at least 12 genes and ≥40% DEGs with the same response. Domains located at a distance ≤5 genes were fused into one and, in addition, clusters that exclusively contained genes belonging to a unique co-transcriptional unit were excluded. To identify topological domains in the transcriptome of strains Δ*topA*P_Zn_*topA*(pLS1ROM*topAWT*) and Δ*topA*P_Zn_*topA*(pLS1ROM*topARA*) in the presence of NOV and in conditions of *topA* induction (SM medium) or non-induction (S medium), we used the criteria described above, which we had established. This allowed the detection of Sc-regulated transcriptional domains, whose size was larger in cultures grown in the presence of SM than in the presence of S (Figure 10a), correlating with the level of TopoI production (Figure 10b). In fact, differences in TopoI expression were detected, being about 3-fold higher in SM media than in S media. However, changes in the genes of the other topoisomerases were equivalent in SM or S media. The distribution of topological domains varied depending of the levels of TopoI expression and the number of DEGs located in UP or DOWN domains increased with respect to those in R6 under physiological conditions. This reached about 37% of the genome in R6 [35], 38% in Δ*topA*P_Zn_*topA*(pLS1ROM*topA*) in the presence of S, and 52% in the presence of SM. The number of DEGs located in UP-domains increased from 293 (R6) to 406/400 (TopoIWT/TopoIRA) in S-grown cultures and 541/544 (TopoIWT/TopoIRA) in SM-grown cultures. The increase in the number of DEGs was higher in DOWN-domains than in UP-domains. It changed from 279 (R6) to 366/375 (WT/RA) in S-treated cultures or 542/506 (TopoIWT/TopoIRA) in SM treated cultures. No significant changes in the transcriptomes of strains overproducing TopoIWT or TopoIRA were found.

## 3. Discussion

The results of this study confirm by means of directed mutagenesis in *S. pneumoniae topA* that TopoI is indeed the target of SCN, as previously anticipated by our group, in both *S. pneumoniae* [12,23] and *M. tuberculosis* [14]. We were previously unable to obtain spontaneous mutants resistant to SCN, which suggests that the alteration of the single type I topoisomerase in *S. pneumoniae* could be lethal, especially if the alteration would be at the DNA-binding site, the site of interaction with SCN. The absence of changes at the DNA-binding site in clinical isolates supports the essentiality of these residues (Figure 1 and Figure 2). In *E. coli*, which possess two Type I topoisomerases (TopoI and TopoIII), a point mutation in TopoI (R168C change, equivalent to R156 of *S. pneumoniae*) located in the catalytic site of the enzyme has been obtained, although the mutant strain shows increased mutation frequencies [36]. However, *E. coli* cells lacking their two type I topoisomerases (TopA and TopB) are not viable [37]. In bacteria that have a single Type I topoisomerase, such as *S. pneumoniae* and *M. tuberculosis*, SCN is highly active [12,13,14]. 

By using directed mutagenesis, we were able to obtain a *topARA* mutant. The mutated residue (R102) is predicted to establish a cation-π interaction with SCN, which suggests that the R102A substitution would confer resistance by interfering this interaction. In accordance, TopoIRA protein showed 20-fold lower specific activity in vitro and 2-fold higher SCN resistance (Figure 3 and Figure 4). Although we were unable to substitute *topAWT* by *topARA* in the chromosome, we tested the in vivo activity of TopoIRA compared to TopoIWT by analyzing genetically constructed strains. We made a strain in which *topARA* and *topAWT* were located at an ectopic location under the control of the controllable P_Zn_ promoter (Figure 5A). Overexpression of TopoI enzymes with respect to R6 strain in the presence of Zn was modest, of about 2-fold. Even with the low expression levels, differences in NOV susceptibility between TopoIRA and TopoIWT overproduction were observed, the strain expressing TopoIRA more susceptible to NOV (Figure 5), which suggests that TopoIRA is less active in vivo than TopoIWT. Measurements of Sc densities in pLS1 topoisomers confirmed this point, since higher relaxation was observed in the presence of NOV when TopoIWT was overexpressed than when it was TopoIRA (Figure 5d). The main source of TopoI in the presence of NOV would be the P_Zn_*topA* copy (either *topAWT* or *topARA*) given the decrease of the transcription of *topAWT* in its natural chromosomal location (a DOWN-regulated domain) [9]. This could provide an explanation for the increased NOV sensibility of the strain carrying the mutant allele, since the specific activity of TopoIRA is 20-fold lower than that of TopoIWT, treatment of this strain with NOV would result in a deficiency in TopoI activity, and impaired growth.

TopoIRA was more resistant to SCN in vivo (Figure 6). This was tested in strains that carried TopoI coding genes under the control of P_MAL_ into recombinant plasmids pLS1ROM-*topAWT* or -*topARA*. These plasmids were introduced in a strain in which *topA* was under the control of P_Zn_ (Figure 6a). Under the growth conditions used in the experiments, the only TopoI enzyme synthesized came from the recombinant plasmids, which carry a 6×—His tail (Figure 6c), which has a lower mobility in the Western blots. The net increase in TopoI enzymes with respect to the wild-type strain R6 was of about 9-fold as detected by Western blot and of about 5-fold (TopoIWT) and 6-fold (TopoIRA) as detected by RNASeq. Sc densities of cultures overexpressing WT or RA enzymes showed significant decreases with respect to no overproduction (Figure 7a). 

The role of TopoIWT and TopoIRA in global transcription was analyzed in cultures overproducing the enzymes. The analysis of the transcriptomic response to TopoIWT and TopoIRA overproduction revealed differences between the DEGs, when the increases were modest (of about 5-fold). However, in the presence of NOV, when the increases were higher (more than 10-fold), no differences in the DEGs were detected between the overproduction of the two enzymes (Figure 8 and Figure 10). The high relaxation reached with the combined effects of NOV (inhibition of gyrase activity) and TopoI overproduction (Figure 9) triggered a transcriptomic response into domains, which was more pronounced than that observed in R6 strain under the same conditions [9]. In fact, DNA relaxation was higher in the strains overproducing TopoI enzymes (−0.010 for TopoIWT and −0.006 for TopoIRA) than in those not overproducing. The high relaxation induced broader domains. This effect was observed in both TopoIWT and TopoIRA overproduction conditions. This transcriptomic pattern could be attributed to the high level of TopoI overproduction, since in R6, under the same NOV treatment, there is a decrease of *topA* transcription of about 23-fold. These results support the organization of the genome into topological domains, which are clearer under extreme DNA relaxation. Genes forming part of domains made up 38% in Δ*topA*P_Zn_*topA* (pLS1ROM*topA*) in the presence of S, consistent with the 37% value in the R6 genome [9,26] under equivalent NOV treatment (10 × MIC, 30 min). These values are in accordance to the Sc density values in those strains under 10 × MIC of NOV: −0.024 (R6) and −0.023 (Δ*topA*P_Zn_*topA* (pLS1ROM*topA*)). However, genes forming part of domains reached 52% of the genome in the presence of SM, i.e., in conditions of TopoI overproduction. These results support the essential role of TopoI in transcription, its suitability as an antibiotic target, and open the possibility to apply combination therapies with antibiotics targeting TopoI, such as SCN, and RNA polymerase inhibitors, such as rifampicin. 

To summarize, this study corroborates that SCN is an inhibitor of TopoI. The *topARA* mutation, located at the DNA binding site in one of the SCN-interacting residues, codes for an enzyme 20-fold less active and 2-fold more resistant to SCN than TopAWT. Comparison of the in vivo activities of strains with regulated expression of *topARA* and *topAWT* showed higher SCN-resistance and lower Sc for TopoIRA. In addition, this study also reinforced the importance of TopoI in the global regulation of transcription. Transcriptome of NOV-treated cells augmented the size of the topological domains, associated to a high DNA relaxation, as estimated by super-resolution confocal microscopy.

## 4. Materials and Methods

### 4.1. Microbiological Methods and Genetic Constructions

*S. pneumoniae* was grown at 37 °C in a casein hydrolysate-based liquid medium (AGCH) containing 0.2% yeast extract and 0.3% sucrose [38]. To obtain *topA* mutants, the Phusion^®^ Site-directed mutagenesis Kit (New England Biolabs, Boston, MA, USA) was used. Three pairs of complementary phosphorylated oligonucleotides were used to amplify the linearized plasmid pQE-*topAWT*-6His [12], which carries the wild-type *topA* with a 6His-tag at the N-terminus. The forward oligonucleotides were 5′-GCGAGTGACCCGGACGCTGAAGGAGAAGCG-3′ (to introduce the R102 (CGT) to A (GCT) change); 5′-GGAAGGTAAACTGGCTGATGTCGAAGTTGG-3′ (to introduce the D543 (GAT) to A (GCT) change); 5′-GGTAAACTGGATGATGTCGCA-3′ (to introduce the E546 (GAA) to A (GCA) change); 5′-GGTAAACTGGATGATGTCAAAGTTGGAAAAGAGCAGTGGCG-3′ (to introduce the E546 (GAA) to A (GCA) change). After amplification, plasmids were circularized by ligation. The plasmid carrying wild-type *topA* was transformed into *E. coli* M15 (pREP4), the plasmids carrying *topA* mutants were transformed into *E. coli* XL1-Blue.

Derivatives of plasmid pLS1ROM [39] carrying *topA* alleles were constructed as follows: fragments of 2149 Kb containing 6-His*topAWT* or 6His-*topARA* were obtained by PCR using Pfu DNA polymerase (Promega), which renders blunt ends, using pQE-6His*topAWT* and pQE-6His*topARA* plasmids as templates, and using primers HisTopAUP (5′-CACACAGAATTCATTAAAGAGG) as forward and TopADown2BamHI ((5′-gcgcgcGGATCCTTATTTAATCTTTTCTTCCTC) as reverse. Restriction enzyme sites are shown underlined in all oligonucleotides. The resulting PCR products were digested with BamHI and ligated with pLS1ROM digested with SmaI and BamHI, resulting in recombinant plasmids pLS1ROM*topAWT* and pLS1ROM*topARA*, which were used to transform R6 competent cells. Transformants were selected by plating in AGCH-agar medium with 0.8% sucrose (to avoid overexpression of *topA*) containing 1 µg/mL erythromycin. pLS1ROM*topAWT* and pLS1ROM*topARA* plasmids were then used to transform strain Δ*topA*P_Zn_*topA*, constructed as previously described [10]. Transformants were selected by plating in AGCH-agar medium with 0.2% yeast extract, 0.8% sucrose, 150 µM ZnSO_4_, and 1 µg/mL erythromycin. 

Strain R6P_Zn_*topAWT* was constructed as described [10]. To construct R6P_Zn_*topARA* a similar procedure was followed, except that the *topARA* allele was amplified from plasmid pQE-SPN*topARA*.

### 4.2. Purification of Proteins 

The pQE1 vector/*E. coli* M15 (pREP4) host cloning system permits the hyperproduction of proteins of genes placed under the control of a phage T5 promoter and two *lac* operator sequences. The host strain contains plasmid pREP4 that constitutively expresses the LacI repressor. Expression of proteins was induced by the addition of isopropyl-β-D-thiogalactoside (IPTG), which binds to the LacI protein and inactivates it. This inactivation allows the host cell’s RNA polymerase to transcribe the genes from the T5 promoter. Cultures of *E. coli* M15 (pREP4)/pQE-*topA* were grown at 37 °C in LB medium containing 250 μg/mL of ampicillin (to select pQE1) and 25 μg/mL kanamycin (to select pREP4) to DO_600_ = 0.6. IPTG (1 mM) was then added and incubation continued for 1 h. Cells were collected by centrifugation, lysed at 4 °C for 1 h in buffer A (Tris-HCl pH 8, 300 mM NaCl, 10 mM imidazole, 1 mM PMSF) containing lysozyme (1 mg/mL) and Triton X-100 (0.2%). They were then sonicated for 20 min (15 pulses of 20 s, with 1 min cooling between each sonication, 100% amplitude) using a Vibra cell sonicator (Sonics, Bristol, CT, USA). Debris was removed by centrifugation at 10,000× *g* at 4 °C for 15 min and the resulting supernatant filtered using a 0.45 µm syringe filter. TopA proteins were purified by affinity chromatography in Ni-NTA (QIAGEN) columns following manufacturer’s instructions as described previously [40]. Eluted proteins were dialyzed overnight at 4 °C against buffer B (50 mM Tris-HCl pH 8, 300 mM NaCl, 1 mM DTT).

### 4.3. Western Blot Assays

Assays were performed as described previously [10]. Briefly, whole cell lysates (~5 × 10^5^ cells) were obtained by centrifugation of 10 mL of culture (OD_620nm_ = 0.4), suspended in 400 µL of phosphate buffered saline, and sonicated. Proteins from lysates (~2 × 10^4^ cells) were separated on polyacrylamide gels, transferred to 0.2 µm PVDF membranes, and incubated with anti-TopoI (diluted 1:500), anti-GyrA (diluted 1:2000) [24], anti-GyrB (diluted 1:2000), and anti-RpoB [32] (diluted 1:500). Molecular masses of RpoB, GyrA, GyrB and TopoI are 134.34 kDa, 92.04 kDa, 72.26 kDa, and 79.38 kDa, respectively. Determinations were performed in triplicate.

### 4.4. Analysis of the Topology of Plasmids

Assays were performed as described previously [10]. Plasmid DNA topoisomers were analyzed in neutral/neutral two-dimensional agarose gels. The first dimension was run at 1.5 V/cm in a 0.4% agarose (Seakem-Lonza, Basel, Switzerland) gel in 1 × Tris-borate-EDTA (TBE) buffer for 20 h at room temperature. The second dimension was run at 7.5 V/cm in 1% agarose gel in 1 × TBE buffer for 9 h at 4 °C for pLS1 plasmid and at 5 V/cm in 0.5% agarose gel for 18 h at 4 °C for pLS1ROM plasmid. Chloroquine (Sigma-Aldrich-Merck, Madrid, Spain) was added to the TBE buffer in both, the agarose and the running buffer. Gels were stained with 0.5 µg/mL ethidium bromide. Images were captured in a ChemiDoc Imaging System and analyzed with the Image Lab software (BioRad, Hercules, CA, USA). 

The DNA linking number (Lk = Tw + Wr) was estimated by quantifying the amount of every topoisomer. The DNA supercoiling density (σ) was calculated using the equation σ = ΔLk/Lk_0_. Changes in the linking number (ΔLk) were determined using the equation Lk = Lk–Lk_0_, in which Lk_0_ = N/10.5, N is the length of the DNA strand in bp, and 10.5 the number of bp per complete turn in B-DNA. To simplify, σ = Lk of the the most abundant topoisomer/(N/10.5). The most abundant topoisomer was identified by densitometry and its Lk was calculated considering that the topoisomer that migrates with ΔLk = 0 in the second dimension has ΔWr = −14 (pLS1), −21 (pLS1ROM) or −27 (pLS1ROM*topA* derivatives), given the positive supercoils introduced by 2µg/mL chloroquine.

### 4.5. Relaxation of pBR322 by TopoI 

Relaxation reactions of pBR322 by TopoI were carried out exactly as described previously [12]. Reactions of 200 µL contained 0.5 µg of CCC pBR322 in 20 mM Tris-HCl pH 8, 100 mM KCl, 10 mM MgCl_2_, 1 mM DTT, 50 µg BSA/mL and TopoI at the indicated concentrations. Incubation with TopoI was at 37 °C during 1 h and reaction was terminated by 2 min incubation at 37 °C with 50 mM EDTA. Treatment with SCN was performed by preincubation of TopoI during 10 min at 4 °C in a final volume of 15 µL. Then, an additional incubation of 1 h at 37 °C with 1% SDS, 100 µg/mL proteinase K was performed. Reaction products were precipitated with ethanol, suspended in electrophoresis loading buffer and analyzed in 1% agarose gels run at 18 V for 18 h. DNA quantification of agarose gels was done by scanning densitometry after electrophoresis and ethidium bromide staining. Quantification of TopoI activity was calculated by gel densitometry using the Image Lab program (Bio-Rad laboratories, Hercules, CA, USA). To calculate activity, the OC and CCC forms amount was determined and divided by the total amount of DNA in each well. IC50 (mean of at least three independent determinations) was defined as the concentration of drug required for a 50% reduction of enzymatic activity.

### 4.6. RNA Extraction and RNA Library Preparation for RNA-Seq

Total RNA from 10 mL of cultures (2 to 4 × 10^8^ cells) was obtained using the RNeasy mini kit (Quiagen, Hilden, Germany). A total of 1 µg of RNA was used for library construction. The Illumina Stranded Total RNA Prep Ligation with Ribo-Zero Plus kit (San Diego, CA, USA) was used, according to the manufacturer’s instructions. RNA was processed to enzymatically remove ribosomal RNA. Briefly, depleted RNA was fragmented and denatured, and cDNA was synthetized. After adenylating 3′ ends and ligating pre-index anchors, anchor-ligated DNA fragments were enriched by 12 cycles of PCR and dual indexes were added to the library. Library quality control was assessed using a 2100 Bioanalyzer Instrument (Agilent, Santa Clara, CA, USA). NovaSeq 6000 Sequencing System was used to sequence the library. 

### 4.7. RNA Seq Data Analysis

Analysis of RNA-Seq data was carried out using the Web-based platform Galaxy. Quality of raw sequence data was analysed with the FASQC tool. Sequencing reads were mapped against the *S. pneumoniae* R6 genome (ASM704v1) using BWA software package (Galaxy version 0.7.17.4) in simple Illumina mode. The number of reads overlapping each coding gene was obtained using program feature Count. Count tables were used as input in DESeq2 for the analysis of differential expression. A threshold *p*-value-adjusted of 0.01 was considered. 

### 4.8. Nucleoid Staining and Confocal Microscopy

We followed our previously described procedure [10], except that nucleoids were stained with 2.5 µM Sytox Green Thermo Fisher (Waltham, MA, USA) instead Sytox™ Orange. Strains were grown to mid-log growth phase, samples (from 5 × 10^7^ to 2 × 10^8^ cells) were collected and about 6 × 10^5^ cells were suspended in buffered salt solution (10 mM Tris, pH 7.6; 137 mM NaCl; 5.4 mM KCl) and stained. Slides were observed using a confocal microscope STELLARIS 8 –FALCON/STED (Leica Microsystems, Wetzlar, Germany) with a HC PL APO 100 ×/1.40 NA × OIL immersion objective. Super resolution images were acquired by Stimulated Emission Depletion (STED) microscopy using 660 nm depletion laser. Image J software was used to analyse the images. Image analysis was performed with Cell profiler v4.2.1 with a customized pipeline to define DNA condensed/relaxed areas per cell. Analysis script first identify the full bacteria then, filters all incomplete or out of focus ones (area 0.125–1.375 µ^2^ and eccentricity below 0.75), condensed DNA was detected using size (below 0.125 µ^2^) and intensity contrast. More than 1800 cells were quantified per sample, relaxed DNA was segmented by subtracting condensed area to total positive region. Ratio of condensed versus relaxed was calculated and graphed using Microsoft Excel software.

## Figures and Tables

**Figure 1 ijms-24-15800-f001:**
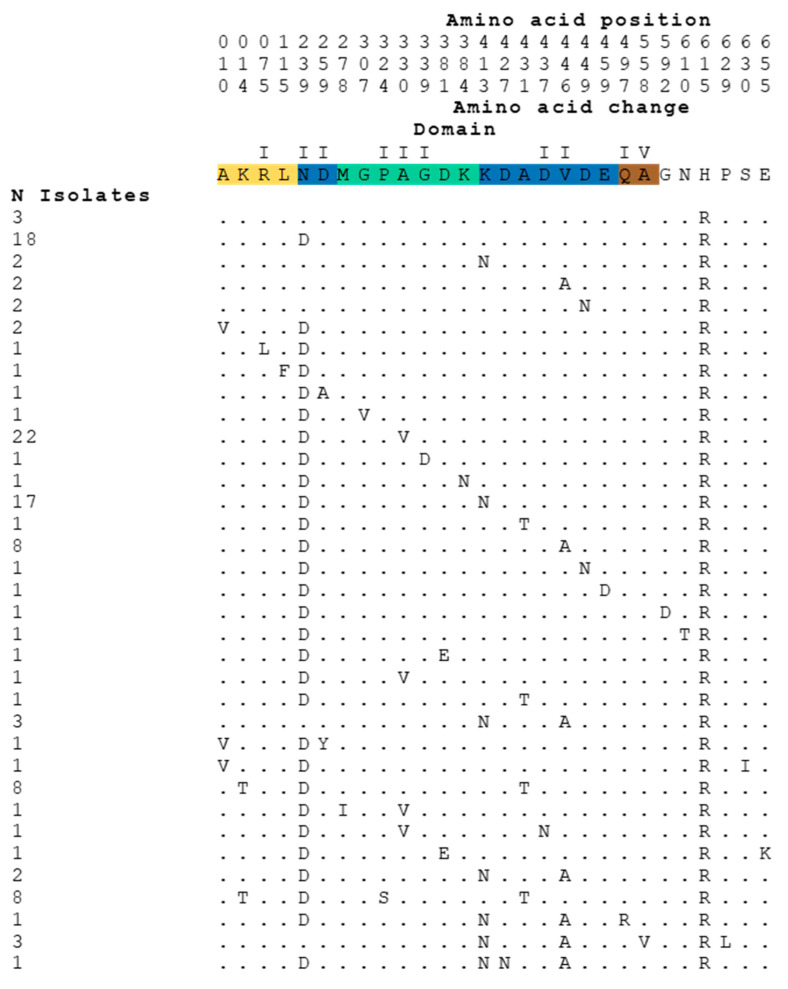
Amino acid changes in TopoI enzymes of *S. pneumoniae* found in 131 sequences of the NCBI data Bank.

**Figure 2 ijms-24-15800-f002:**
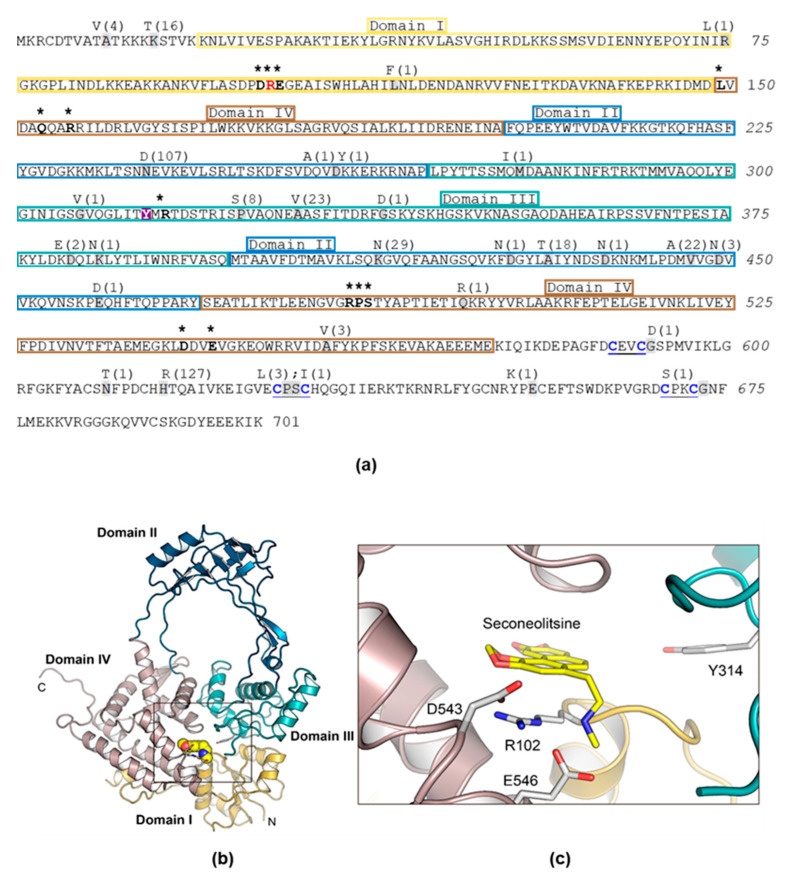
Primary structure of *S. pneumoniae* TopoI, structural model, and modelling of the interaction with SCN. (**a**) Amino acid sequence showing structural domains illustrated in (**b**) and residues involved in nucleotide binding (written in boldface and labelled with asterisks). The catalytic Y-314 residue is shadowed in magenta. The R-102 residue is shown in red. Grey shadowed residues are those that varied in 137 clinical isolates accessible in the nucleotide databases, and letters above them are the residue changes observed in the number of isolates indicated in parentheses. (**b**) Overall structure of the 67 kDa fragment showing the four structural domains indicated in (**a**). (**c**) SCN bound to the nucleotide-binding site of TopoI. The residues forming the binding site are drawn as capped sticks.

**Figure 3 ijms-24-15800-f003:**
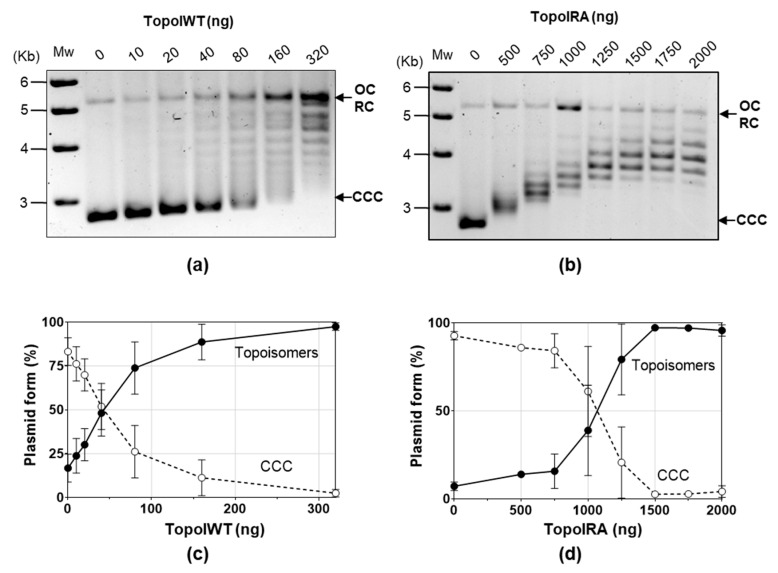
TopoIRA is less active than TopoIWT. (**a**,**b**) Plasmid pBR322 (0.5 µg) was incubated with the indicated amounts of purified TopoI enzymes for 1 h at 37 °C. Mw, size of DNA markers. (**c**,**d**) Quantification of TopoI activity determined as increase of topoisomers (including OC/RC forms) and decrease of CCC form (mean ± SEM, *n* = 3).

**Figure 4 ijms-24-15800-f004:**
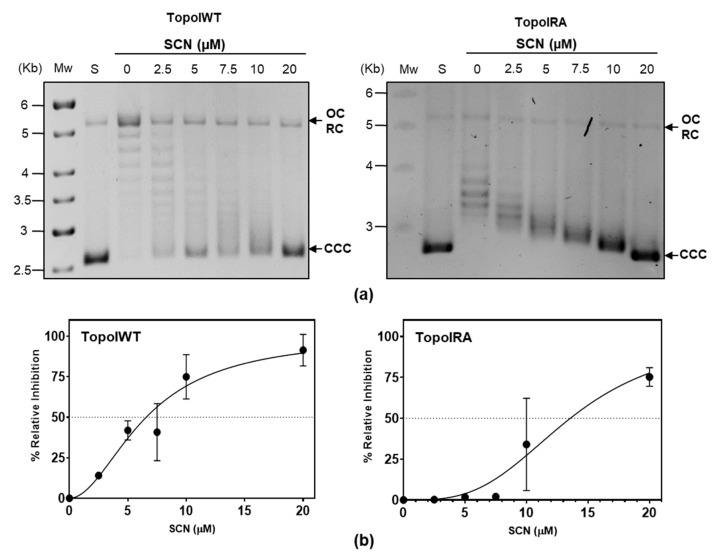
TopoIRA showed increased SCN resistance in vitro. (**a**) pBR322 was treated with 1 unit of purified enzymes in reactions containing SCN at the concentrations indicated. S, pBR322 used as a substrate; Mw, molecular DNA marker size. Symbols: CCC, covalently closed circles; OC, relaxed open circles. (**b**) Quantification of the activity of TopoI determined as in Figure 3c.

**Figure 5 ijms-24-15800-f005:**
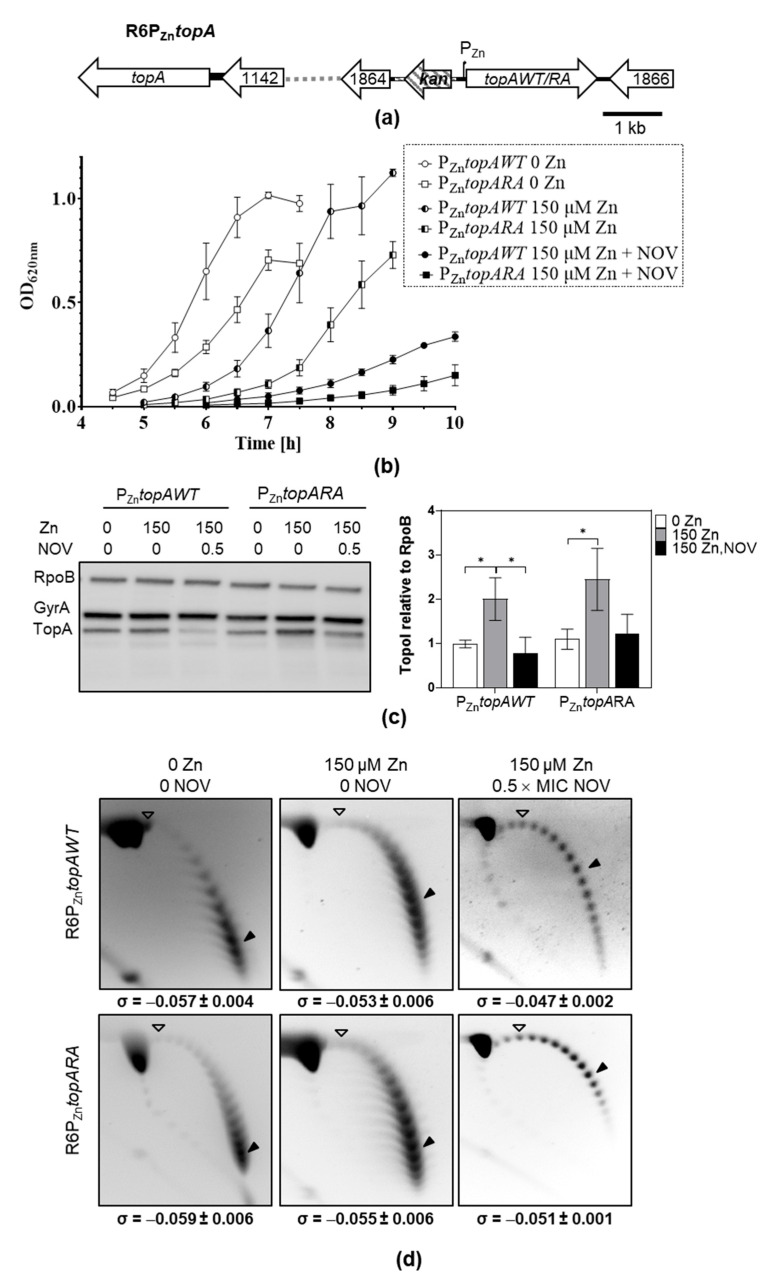
Overproduction of TopoIRA increases lethality and NOV susceptibility. (**a**) Genetic organization of strain R6P_Zn_*topA* in which P_Zn_*topAWT* or P_Zn_*topARA* fusions are located to *spr1865*. (**b**) Growth (mean ± SEM, *n* = 3) of strains. Overnight cultures in AGCH medium without ZnSO_4_ were diluted 1000-fold and grew under the indicated conditions. Values after reaching the stationary phase were excluded from the graphic for more clarity. (**c**) Western blot analysis of TopoI and GyrA relative to RpoB amount. After dilution, cultures were grown to OD_620nm_ = 0.4, and samples containing 0.12 units of OD_620nm_ were separated by SDS-PAGE, blotted, and membranes incubated with polyclonal anti-GyrA, anti-TopoI and anti-RpoB antibodies. A representative blot is showed together with the quantification of the amount of TopoI relative to RpoB (mean ± SD, *n* = 3). Statistical significance two-tailed Student’s *t*-test, * *p* < 0.05. (**d**) Distribution of topoisomers of plasmid pLS1 from the strains and growth conditions indicated. Samples were run in agarose gels in the presence of 1 and 2 µg/mL chloroquine in the first and second dimensions, respectively. A white arrowhead indicates the topoisomer that migrated with ΔLk = 0 in the second dimension and had a ΔWr = −14 (the number of positive supercoils introduced by 2 µg/mL chloroquine). A black arrowhead indicates the most abundant topoisomer. The corresponding σ value (mean ± SD, *n* = 3) is indicated below each panel.

**Figure 6 ijms-24-15800-f006:**
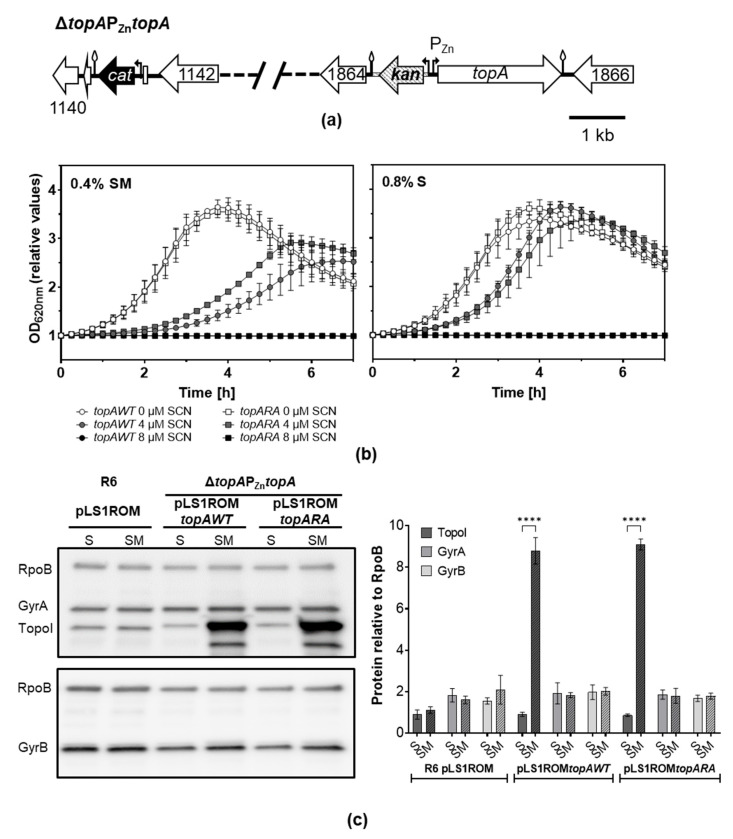
Overproduction of TopoIRA increases SCN resistance. (**a**) Genetic map of strain Δ*topA*P_Zn_*topA* where the chromosomal *topA* gene was deleted by replacement with a cat cassette and a P_Zn_*topA* fusion is located at spr1865. (**b**) Growth of strains Δ*topA*P_Zn_*topA* carrying pLS1ROM*topAWT* or pLS1ROM*topARA* on medium containing S or SM in the presence or absence of SCN. Values of OD_620nm_ relative to time 0 are represented. Overnight cultures grown to OD_620nm_ = 0.4 on medium depleted of ZnSO_4_ containing S were diluted 1000-fold in the indicated media without ZnSO_4_. Growth was monitored in a TECAN Infinite 200 PRO reader. Results shown are the mean ± SEM, *n* = 3. (**c**) Western blot analysis of TopoI, GyrA, and GyrB, relative to RpoB. Cultures were grown as in (**a**) to OD_620nm_ = 0.4 (relative OD_620nm_ value of 2.5) and proteins from samples containing 0.12 units of OD_620nm_ were separated by SDS-PAGE and blotted. The membranes were incubated with polyclonal anti-GyrA, anti-TopoI, anti-GyrB and anti-RpoB antibodies. A representative blot is shown with the quantification of the amount of the proteins relative to that of RpoB (mean ± SD, *n* = 3). Statistical significance two-tailed Student’s *t*-test, **** *p* < 0.0001.

**Figure 7 ijms-24-15800-f007:**
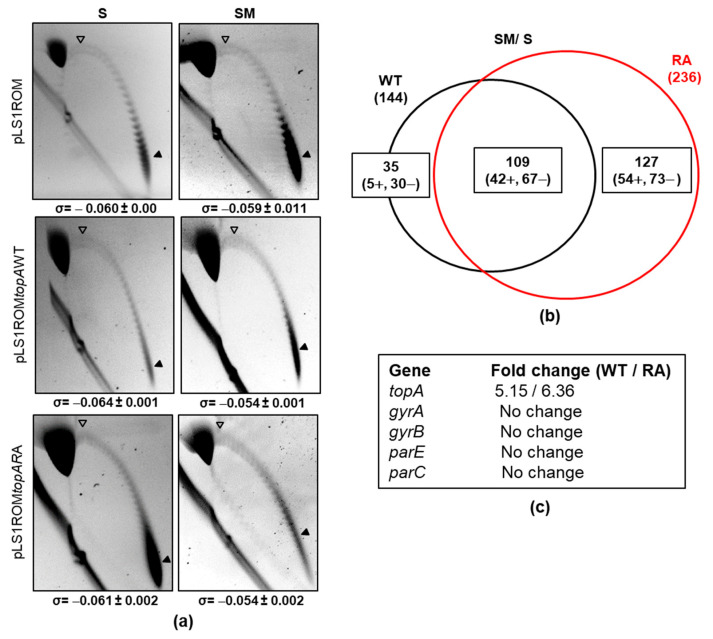
Transcriptomic response and Sc in strain Δ*topA*P_Zn_*topA* carrying pLS1ROM*topAWT* or pLS1ROM*topARA* under overproduction of TopoI enzymes. (**a**) Distribution of topoisomers of plasmids pLS1ROM, pLS1ROM*topAWT* or pLS1ROM*topARA* under non-induction (S) or induction (SM) culture conditions grown as indicated in Figure 6. Samples were taken at OD_620nm_ = 0.4 (relative OD_620nm_ value of 2.5) and plasmids were run in agarose gels in the presence of 1 and 2 µg/mL chloroquine in the first and second dimensions, respectively. White arrowheads indicate the topoisomer that migrated with ΔLk = 0 in the second dimension. Therefore, this topoisomer migrated with ΔWr = −21 for pLS1ROM and −27 for the recombinant derivatives coding the TopoI enzymes. Black arrowheads indicate the most abundant topoisomer. The σ value (average ± SD, *n* = 3) is indicated. (**b**) Venn diagram showing DEGs upon treatment with SM. (**c**) Variations in the expression of genes coding DNA topoisomerases.

**Figure 8 ijms-24-15800-f008:**
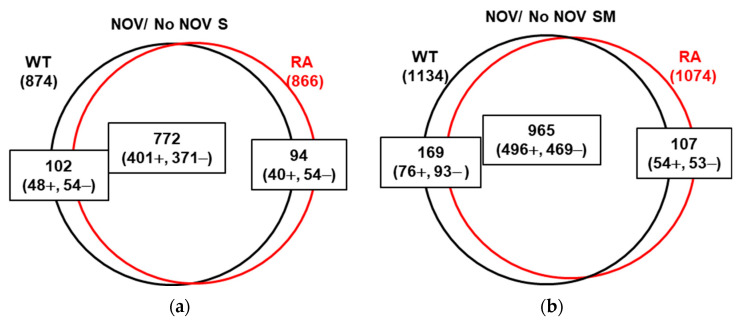
Transcriptomic response to NOV treatment in strain Δ*topA*P_Zn_*topA* expressing *topAWT* or *topARA.* Venn diagram showing DEGs upon treatment with NOV (10 × MIC, 30 min) growing in S (**a**) or SM (**b**) media.

**Figure 9 ijms-24-15800-f009:**
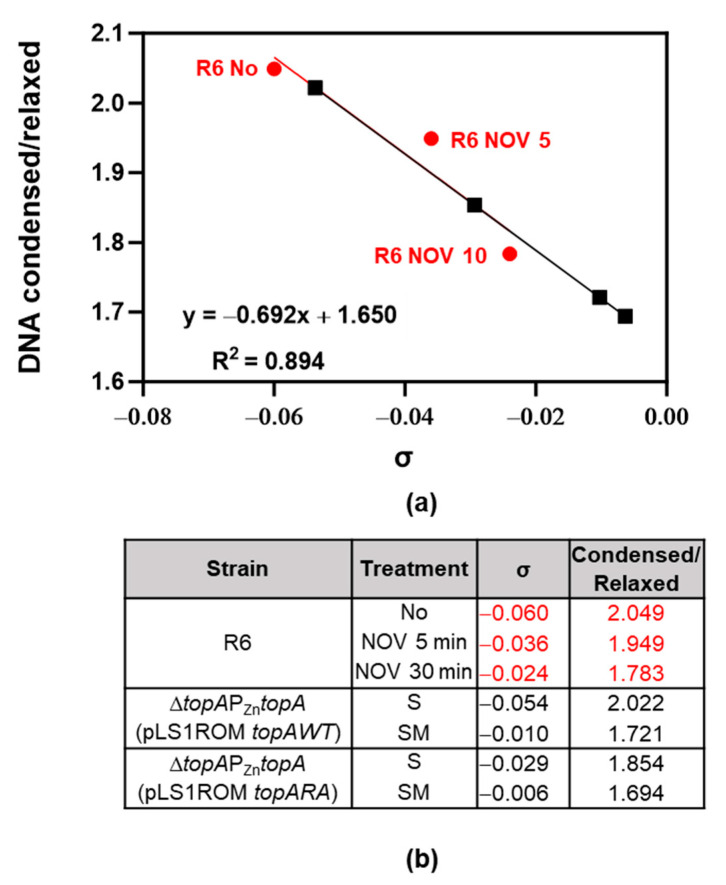
Nucleoid compaction measurement by using super-resolution confocal microscopy. Bacterial DNA was stained by using Sytox Green, image analysis was performed with Cell profiler v4.2.1 with a customized pipeline to define DNA condensed/relaxed areas per cell. More than 1800 cells were quantified per sample. Ratio of condensed versus relaxed was calculated and graphed using Microsoft Excel software. (**a**) Correlation between values obtained by confocal microscopy vs. σ values estimated in 2D-agarose gels of strains and treatment indicated in (**b**). Red dots and red numbers are controls, black squares and black numbers are unknown values.

**Figure 10 ijms-24-15800-f010:**
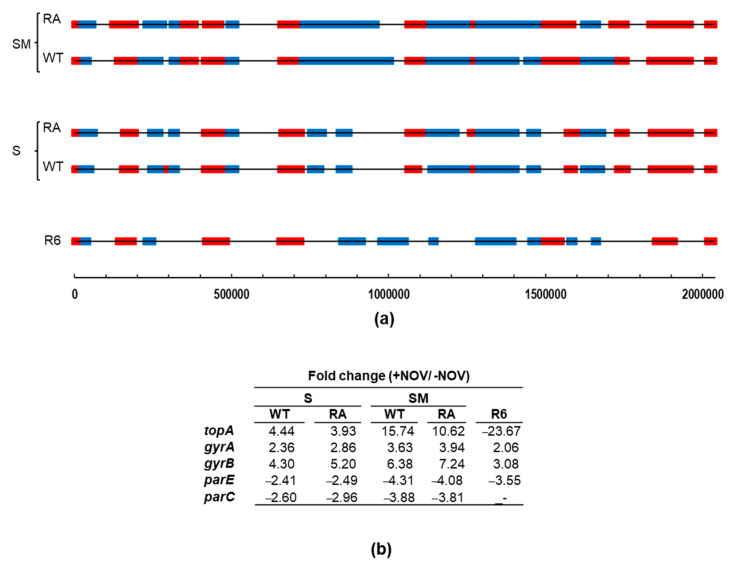
Transcriptomic response after NOV treatment of strains overproducing TopoI. (**a**) Location of topological domains containing UP-regulated (red) or DOWN-regulated (blue) DEGs after NOV treatment (30 min 10 × MIC) in strains R6 (as taken from [9]), Δ*topA*P_Zn_*topA* carrying pLS1ROM*topAWT* (WT), or pLS1ROM*topARA* (RA). Strains were grown under induction (SM medium) or non-induction (S medium) conditions. DEGs showing a fold change ≥ 2 (absolute value) and a *p*-value-adjusted ≤ 0.01 were considered. DEGs are represented against the 5′ location of each open reading frame in R6. (**b**) Variations in the expression of genes coding DNA topoisomerases.

## Data Availability

The data discussed in this publication have been deposited in NCBI’s Gene Expression Omnibus and are accessible through GEO Series accession number GSE243041 (https://www.ncbi.nlm.nih.gov/geo/query/acc.cgi?acc=GSE243041, accessed on 26 October 2013). The following secure token has been created to allow review of record GSE243041 while it remains in private status: ibazowcanhqjvsz.

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
