# Peer review of "Physiologic and Transcriptomic Effects Triggered by Overexpression of Wild Type and Mutant DNA Topoisomerase I in Streptococcus pneumoniae"

_ijms, 2023, doi:10.3390/ijms242115800_

Round 1

Reviewer 1 Report

Comments and Suggestions for Authors

García-López et al. in their paper titled “Physiologic and transcriptomic effects triggered by overexpression of wild type and a DNA topoisomerase I mutant in Streptococcus pneumoniae.”describe further extension of their studies on sole TopoisomeraseI (topoI) of Streptococcus pneumoniae. By employing the overexpression strains of both wild type (wt) and a mutant of topoI(mutation DNA binding/ in one of the SCN-interacting residue having resistance to  SCN ) they have carried out comparative transcriptomic studies. This study further  substantiates that SCN is a inhibitor topoI and the enzyme’s importance  in the global regulation of transcription in this clinically important organism. In addition transcriptome of Novobiocin treated cells suggest that higher relaxation obtained under drug treated condition leads to broadening of topological domains in this organism. This yet another excellent study from Adela G. de la Campa’s group on topoisomerases from Streptococcus pneumoniae.

The paper should be published. It is an important contribution un further understanding the role of topoI in transcription regulation.  However, its current form has too many manuscript organisational issues. The manuscript is poorly written. The data can be nicely presented with improved writing.  The following points should improve the presentation of the manuscript.

a)The abstract can be expressed more effectively as the message is not clearly depicted. For example, in line 18,19,20: The sentence is too long and does not provide required message. It can be rephrased as- “We compared the in vivo activity of TopoIRA and TopoIWT using tuneable or regulated overexpression strains”.

b)Introduction is not sequential and not  cohesive.   Each paragraph describes different aspects of topoI with minimal connectivity to the each other. Sentences are too long and confusing. For better readability and understanding, it is imperative that the authors carefully prepare the manuscript.

c)Results section 2.2“Role of TopoIR102A in Cell Viability and NOV Resistance in vivo”. Difference in optical density does not necessarily reflect their viability status. CFU studies will be more conclusive in this regard.

d)It is often stated in the manuscript that change in “global” transcription upon topoI overexpression. To support, the RNA-Seq data can be better presented taking more examples of global level changes of genes in the results section.

e)I am indicating a few typographical and sentence corrections. There far too many which needs to be corrected-

Line 19 -  TopoIRA in vivo analyzing strains that; Correct it as -   TopoIRA in vivo, by analyzing….

Line 36-38  the way written, do not make sense. Should read as “Resistance of the bacterium to beta-lactams and macrolides has spread, which has resulted in recommending fluoroquinolones, which target type II topoisomerases, for treatment. Similarly Line 61, sentence needs correction. Does not make sense as written.

Additional points/corrections-

Scientific names and in “vivo/in vitro” should be italicized throughout the paper.

Line 38: Which cleave both strands of the DNA transiently,

Fig 2d,e, what is y-axis (ng) needs to be mentioned.

Line 382, “difficulting” word should be replaced.

Line 386: “genetic constructed strain” may be written as genetically constructed/ engineered strain.

Line 388: absence of “to” after “with respect”.

Line 54:Which “other topoI inhibitors”? You may refer/cite Godbole et al 2014 and 2015.

Line 58: ‘and an increase in gyrase genes’!! Change to ‘and an increase in transcription of gyrase genes’.

Line 61: “being HU and StaR involved in Sc regulation”. This does not connect to the previous sentence well. Should be rephrased.

Line 62,63:Sentence repetitive to initial sentence of same para.

Line 64,65,66,67: Consider rephrasing.

Paragraph from 56-79: The connection of this information with the rest of the paper is not clear. Also, sentences are too long and not reader friendly.

Line 106: ‘with respect the sequence’  Insert ‘to’ to read as –‘with respect to the sequence’

Line 116-117:perhaps because it does not form part of the DNA-binding site.It would be better to mention Gate strand binding site. Domain II otherwise shows interaction with T-strand.

Line 132:Cloned into plasmid?E. coli should be italic.

Line 136: E. coli should be italic (similar mistakes at many places)

Line 138:Site directed mutagenesis is a precise technique. How did it create internal stop codon mutations?

Line 138-142: Fits better in  materials and methods.

Line 143,144:WTand RA should be replaced with TopoIWT/ TopoIRA to maintain consistency.

Line 147-150:Is the resistance is as a result of mutation or due to amount of enzyme per unit? The comparison is not very clear. If the amount of enzyme per unit is different in TopoIWT and TopoIRA, this will require different amounts of SCN to inhibit the reaction. It would be better to consider the enzyme : SCN ratio to inhibit the reaction and then compare between TopoIWT and TopoIRA.

Figure2 legend: E.coli and gene names should be italics

Fig 4b,c:The western blot shows that in presence of 150uM Zn the levels of ectopic TopoI are not significantly increased compared to 0uM Zn. Why is the growth of R6PzntopAWT effected in presence of Zn with respect to R6PzntopAWTin absence of Zn?

Fig 4c:The strains used in the experiment are similar to the strains from reference 10. The induction with 150uM concentration of Zn shows approximately 3 fold increase in TopoI levels as per the reference 10. However, in figure4c of this manuscript same results are not observed even though the experimental conditions mentioned are same.

Fig 4b vs Fig 5b: In 4b the overexpression of TopoIRA reduces the growth of S. pneumonia compared to TopoIWT overexpression. The same result is not seen in Fig 5b where the overproduction of TopoIRA is expected to result in enhanced growth defect (for the control cultures, 0uM SCN).

Figure 5b: The resistance to SCN at 4uM concentration is not consistent in moderate expression (S) vs overexpression (SM) for TopoIWT/RA. There is no explanation provided by authors.

Results, 2.4 and 2.5 sections need to be re-written. The sentences are confusing with a number of mistakes. The data is not explained sequentially. For example line 337-340To identify topological domains in the transcriptome of strains DtopAPZn-topA(pLS1ROMtopAWT) and DtopAPZn-topA(pLS1ROMtopARA) in the presence of NOV  and in conditions of topA induction (SM medium) or non-induction (S medium), we used the criteria previously established by us [24] to identify topological domains.” The sentence is lengthy with repetition ‘identify topological domains’. There is no prior explanation for what topological domains mean.

Line372: Correct it as “the site of interaction with SCN”

Line 377-379: How is essentiality of E.coli TopoI/III genes related to SCN potency against M. tuberculosis? Please split the sentence. Also reference 14 is not appropriate to cite the presence of sole topoI in M.tuberculosis. Please cite Ahmed et al 2014 or reference within with respect  M.tuberculosis topoI. Also note SCN potency is not comparable the specific inhibitors of mycobacterial topo I inhibitors.

Line378: Change to ‘has been proven’

Line 380-383:“Conferring resistance” is repetitive

Discussion has lots of mistakes. Consider re-writing.

4.2. Western Blot Assays: All details are copy pasted from reference 10. Please cite the reference.

Comments on the Quality of English Language

It is included in the above section. A lot of suggestions are provided

Author Response

We are glad that you consider our study as excellent. Thank you for the careful corrections on the manuscript that have been of great help to improve it. We have answered point by point to your concerns.

a)The abstract can be expressed more effectively as the message is not clearly depicted. For example, in line 18,19,20: The sentence is too long and does not provide required message. It can be rephrased as- “We compared the in vivo activity of TopoIRA and TopoIWT using tuneable or regulated overexpression strains”.

Following your suggestions, the abstract has been rewritten.

b)Introduction is not sequential and not  cohesive.   Each paragraph describes different aspects of topoI with minimal connectivity to the each other. Sentences are too long and confusing. For better readability and understanding, it is imperative that the authors carefully prepare the manuscript.

We have significantly reduced the introduction significantly, from 884 to 684 words, about 23%. We have tried to connect the sentences and, in addition, we have added some sentences describing the main motivation of the study in the last paragraph of the introduction.

c)Results section 2.2“Role of TopoIR102A in Cell Viability and NOV Resistance in vivo”. Difference in optical density does not necessarily reflect their viability status. CFU studies will be more conclusive in this regard.

We agree that determination of CFUs would be more informative; however, our OD determinations clearly show the differences in growth between the strains overproducing TopoIWT and those overproducing TopoIRA.

d)It is often stated in the manuscript that change in “global” transcription upon topoI overexpression. To support, the RNA-Seq data can be better presented taking more examples of global level changes of genes in the results section.

We considered that the results of the global transcription data are well represented through Venn diagrams (Figures 6 and 7) and the characterization of the topological domains (Figure 9). In addition, all data have been deposited in NCBI's Gene Expression Omnibus and are accessible  through  GEO  Series  accession  number    GSE243041  (https://www.ncbi.nlm.nih.gov/geo/query/acc.cgi?acc=GSE243041). The following secure token has been created to allow review of record GSE243041 while it remains in private status: ibazowcanhqjvsz.

e)I am indicating a few typographical and sentence corrections. There far too many which needs to be corrected-

Line 19 -  TopoIRA in vivo analyzing strains that; Correct it as -   TopoIRA in vivo, by analyzing….

Corrections have been made as suggested.

Line 36-38  the way written, do not make sense. Should read as “Resistance of the bacterium to beta-lactams and macrolides has spread, which has resulted in recommending fluoroquinolones, which target type II topoisomerases, for treatment. Similarly Line 61, sentence needs correction. Does not make sense as written.

Corrections have been made as suggested.

Additional points/corrections-

Scientific names and in “vivo/in vitro” should be italicized throughout the paper.

Corrections have been made as suggested.

Line 38: Which cleave both strands of the DNA transiently, OK

Fig 2d,e, what is y-axis (ng) needs to be mentioned. Thank you for the observation.

The x-axis indicates now the amount (in ng) of enzyme used.

Line 382, “difficulting” word should be replaced. OK

Line 386: “genetic constructed strain” may be written as genetically constructed/ engineered strain. OK

Line 388: absence of “to” after “with respect”.

Corrections have been made as suggested.

Line 54:Which “other topoI inhibitors”? You may refer/cite Godbole et al 2014 and 2015.

This sentence has been deleted when the introduction has been shortened.

Line 58: ‘and an increase in gyrase genes’!! Change to ‘and an increase in transcription of gyrase genes’. OK

Line 61: “being HU and StaR involved in Sc regulation”. This does not connect to the previous sentence well. Should be rephrased.

It has been rephrased

Line 64,65,66,67: Consider rephrasing. It has been rephrased

Paragraph from 56-79: The connection of this information with the rest of the paper is not clear. Also, sentences are too long and not reader friendly.

It has been rephrased

Line 106: ‘with respect the sequence’  Insert ‘to’ to read as –‘with respect to the sequence’  OK

Line 116-117:perhaps because it does not form part of the DNA-binding site.It would be better to mention Gate strand binding site. Domain II otherwise shows interaction with T-strand.

OK

Line 132:Cloned into plasmid?E. coli should be italic.

OK

Line 136: E. coli should be italic (similar mistakes at many places)

OK, it has been mended

Line 138:Site directed mutagenesis is a precise technique. How did it create internal stop codon mutations?

We do not have an explanation for this, it could be related with the essentiality of topA and its toxicity when overproduced. 

Line 138-142: Fits better in  materials and methods.

OK, it has been moved to material and methods

Line 143,144:WTand RA should be replaced with TopoIWT/ TopoIRA to maintain consistency.

Corrections have been made as suggested.

Line 147-150:Is the resistance is as a result of mutation or due to amount of enzyme per unit? The comparison is not very clear. If the amount of enzyme per unit is different in TopoIWT and TopoIRA, this will require different amounts of SCN to inhibit the reaction. It would be better to consider the enzyme : SCN ratio to inhibit the reaction and then compare between TopoIWT and TopoIRA.

We have calculated the amount of TopoIWT enzyme yielding 50% activity as about 54.4 ng (Figure 2a, c), while it was about 1084 ng for the TopoIRA enzyme (Figure 2b, d). This means that the activity of TopoIRA is 20-fold lower than that of TopoIWT. Then inhibition experiments were performed, as usually, with enzymes at 50% activity (Figure 3), the part of the enzyme activity curve that is more sensitive to changes. These experiments showed that SCN at a concentration of 6.8 µM reduced the activity of the TopoIWT enzyme to 50%, while a concentration of 13.4 µM of SCN was necessary for the same reduction of the TopoIRA enzyme activity, indicating that TopoIRA is 2-times more resistant to SCN inhibition. We cannot consider the ratio as a factor of resistance, given that the compound does not necessarily bind with the same affinity to TopoIWT or TopoIRA.

Figure2 legend: E.coli and gene names should be italics.

Figure 2a has been transferred to supplementary material as suggested by another reviewer. The correction has been made in this supplementary Figure 1.

Fig 4b,c:The western blot shows that in presence of 150uM Zn the levels of ectopic TopoI are not significantly increased compared to 0uM Zn.

Although not clearly visible in the Western image, quantification of three replicates yielded TopoIWT values of 1.0±0.1 ng at 0 µM Zn and 2.0±0.5 ng at 150 µM Zn (2-fold, P = 0,02), as shown in the graph.

Why is the growth of R6PzntopAWT effected in presence of Zn with respect to R6PzntopAWTin absence of Zn?ç

Differences in growth could be due to the high amount of Zn in the medium and the toxic effect of high Zn concentrations.

Fig 4c:The strains used in the experiment are similar to the strains from reference 10. The induction with 150 uM concentration of Zn shows approximately 3 fold increase in TopoI levels as per the reference 10. However, in figure4c of this manuscript same results are not observed even though the experimental conditions mentioned are same.

As you notice, the strain used (PZntopA) in reference 10 is the same. While in this study we have observed a 2-fold increase, in the study of reference 10 (Table 2) the increase of TopoI is 2.1 (25.7 ng at 150 µM and 12.2 ng at 0 µM Zn), equivalent to that observed in this study. Maybe you have considered the value with 300 µM Zn, which was 2.9-fold (35.9 ng versus 12.2 ng at 0 µM Zn). This 300 µM Zn concentration has not been tested in this study.

Fig 4b vs Fig 5b: In 4b the overexpression of TopoIRA reduces the growth of S. pneumonia compared to TopoIWT overexpression. The same result is not seen in Fig 5b where the overproduction of TopoIRA is expected to result in enhanced growth defect (for the control cultures, 0uM SCN).

A putative explanation is that, in Fig 4, the strain PZntopARA produced two different TopoI enzymes, the TopoIWT from the topA gene in its chromosomal position, and TopoIRA under PZn control. These different alleles produced equivalent amounts of enzymes under 150 µM of Zn induction (Figure 4c). The putative competition between TopoIWT and TopoIRA would then be the cause of the negative effect observed on growth.  In agreement, in Figure 5b, the deleterious effect of TopoIRA production is not observed as strain DtopAPZn-topApLS1ROMtopARA did not produce TopoIWT.

Figure 5b: The resistance to SCN at 4uM concentration is not consistent in moderate expression (S) vs overexpression (SM) for TopoIWT/RA. There is no explanation provided by authors.

You are right, but it is important to notice that 4 µM SCN is a subinhibitory concentration, and the effects on growth could be variable depending on the amount of enzyme.

Results, 2.4 and 2.5 sections need to be re-written. The sentences are confusing with a number of mistakes. The data is not explained sequentially. For example line 337-340To identify topological domains in the transcriptome of strains DtopAPZn-topA(pLS1ROMtopAWT) and DtopAPZn-topA(pLS1ROMtopARA) in the presence of NOV  and in conditions of topA induction (SM medium) or non-induction (S medium), we used the criteria previously established by us [24] to identify topological domains.” The sentence is lengthy with repetition ‘identify topological domains’. There is no prior explanation for what topological domains mean.

The last paragraph of section 2.5 has been changed.

Line372: Correct it as “the site of interaction with SCN”

OK

Line 377-379: How is essentiality of E.coli TopoI/III genes related to SCN potency against M. tuberculosis? Please split the sentence. Also reference 14 is not appropriate to cite the presence of sole topoI in M.tuberculosis. Please cite Ahmed et al 2014 or reference within with respect  M.tuberculosis topoI. Also note SCN potency is not comparable the specific inhibitors of mycobacterial topo I inhibitors.

Sorry, what we wanted to reference that SCN is very active in M. tuberculosis [14]. This part has been rephrased.

Line378: Change to ‘has been proven’

OK, it has been changed

Line 380-383:“Conferring resistance” is repetitive

OK, it has been changed

Discussion has lots of mistakes. Consider re-writing.

4.2. Western Blot Assays: All details are copy pasted from reference 10. Please cite the reference.

This has been done

Comments on the Quality of English Language

It is included in the above section. A lot of suggestions are provided

English has been corrected by Pedro A. Lazo-Zbikowski (Instituto de Biologia Molecular y Celular del Cáncer, CSIC, Salamanca, Spain).

Reviewer 2 Report

Comments and Suggestions for Authors

Major Comments:

1. In Table 1- what were the selection criteria for the strains being analyzed? There are many more pneumococcal strains that have been sequenced, or does this subset capture the majority of the genetic variation at this locus?

Minor comments:

1. There are too many details in the abstract (i.e. p-values, fold expression, and relaxation values)- these should be cut to make the abstract more concise.

2. Figure 2A could readily be made into a supplemental figure.

3. Figure 4A- why are the non-induced growth curves cut off at 7.5 hours? Is this because they begin undergoing autolysis and the authors did not want to complicate the graph. If so, this deserves a mention in the results for why this was done.

4. Figure 4D- a better description of what this assay is measuring so that a general reader an interpret the images would have been helpful.

5. Figure 6D could readily be moved into supplemental or eliminated for brevity.

6. Figure 7 could be moved into the supplemental data.

Comments on the Quality of English Language

Overall was quite good throughout. There are a few instances where it could be cut down to be more concise. 

Author Response

Thank you for the careful corrections on the manuscript that have been of great help to improve it. We have answered point by point to your concerns.

English has been corrected by Pedro A. Lazo-Zbikowski (Instituto de Biologia Molecular y Celular del Cáncer, CSIC, Salamanca, Spain).

  1. In Table 1- what were the selection criteria for the strains being analyzed? There are many more pneumococcal strains that have been sequenced, or does this subset capture the majority of the genetic variation at this locus?

The following explanation has been added at the beginning of the 2.1 section of Results:

A BLAST was made with R6 TopoI against the S. pneumoniae sequences available in the NCBI data bank (accessed on 3/18/2022). Among the first 200 sequences producing significant alignments, 131, which had a complete TopA and GyrA sequences, were selected for the analysis.

Minor comments:

  1. There are too many details in the abstract (i.e. p-values, fold expression, and relaxation values)- these should be cut to make the abstract more concise.

Thank you for the suggestion. Most of these data has been removed for clarity in the new version of the abstract.

  1. Figure 2A could readily be made into a supplemental figure.

 Ok, it has been moved to a new Supplementary Material section as Figure S1.

  1. Figure 4A- why are the non-induced growth curves cut off at 7.5 hours? Is this because they begin undergoing autolysis and the authors did not want to complicate the graph. If so, this deserves a mention in the results for why this was done.

The following sentence has been added to Figure 4 (b) legend: Values after reaching the stationary phase were excluded from the graph for more clarity.

  1. Figure 4D- a better description of what this assay is measuring so that a general reader an interpret the images would have been helpful.

Section 4.3 of Material and Methods has been amended for clarity. The following paragraph has been added:

“The DNA linking number (Lk) was calculated by quantifying the amount of every topoisomer. The DNA supercoiling density (σ) was calculated using the equation σ = DLk/ Lk0. Changes in the linking number (DLk) were determined using the equation Lk = Lk -  Lk0, in which Lk0 = N/10.5, N is the length of the DNA strand in bp, and 10.5 the number of bp per one complete turn in B-DNA. To simplify, σ = mode/ Lk0, being mode the whrite of the most abundant topoisomer.

In addition, we have added the following sentence in section 2.2 of the Results: “This estimation of Sc density in the plasmid correlates with nucleoid compaction levels estimated by super-resolution confocal microscopy [10]”.

  1. Figure 6D could readily be moved into supplemental or eliminated for brevity.

 OK, we have moved it to the new Supplementary Material section as Figure S2.

  1. Figure 7 could be moved into the supplemental data.

We have moved Figure 7b to the new Supplementary Material section as Figure S3. We consider the information in Figure 7A to be relevant and we prefer to preserve it.

Reviewer 3 Report

Comments and Suggestions for Authors

Dear Editor,

The manuscript of García-López et al. deals with the investigation at molecular level of the mechanisms of bacterial inhibition by the antibiotic seconeolitsine. Such studies are highly motivated in the light of the constant spread of resistances to antibiotics and the need for a search of new antibiotics. These considerations motivate me to advise you to consider the publication of the manuscript, however, after some major revisions.

Major remarks

1.       The Introduction section should be shortened at least by 30-40%, some of the statements being widely known scientific facts – especially the part on the role of the enzymes. The main motivation of the study, as well as its objectives fade within so much information.

2.       Again, concerning the Introduction section, 2-3 sentences describing the logic of the study and the experimental pipeline would be a great addition.

3.       A serious drawback of the manuscript is that it lacks a conclusion section.

Minor remarks

1.       I would like to suggest using the full name of the antibiotics seconeolitsine and  novobiocin instead of SCN and NOV, the number of the abbreviations used within the manuscript being large enough.

2. Lines 298 and 318 – species name should be in italics.

Author Response

Thank you for the careful corrections on the manuscript that have been of great help to improve it. We have answered point by point to your concerns.

Major remarks

  1. The Introduction section should be shortened at least by 30-40%, some of the statements being widely known scientific facts – especially the part on the role of the enzymes. The main motivation of the study, as well as its objectives fade within so much information.

Thank you for the suggestion. We have reduced the introduction significantly, from 884 to 684 words, about 23%.

  1. Again, concerning the Introduction section, 2-3 sentences describing the logic of the study and the experimental pipeline would be a great addition.

We have introduce some sentences describing the main motivation of the study in the last paragraph of the introduction.

  1. A serious drawback of the manuscript is that it lacks a conclusion section.

Thanks for the suggestion; we have now added the following paragraph:

“To summarize, this study corroborates that SCN is an inhibitor of TopoI. The topARA mutation, located at the DNA binding site in one of the SCN-interacting residues, codes for an enzyme 20-fold less active and 2–fold more resistant to SCN than TopAWT. Comparison of the in vivo activities of strains with regulated expression of topARA and topAWT showed higher SCN-resistance and lower Sc for TopoIRA. In addition, this study also re-inforced the importance of TopoI in the global regulation of transcription. Transcriptome of NOV-treated cells augmented the size of the topological domains, associated to a high DNA relaxation, as estimated by super-resolution confocal microscopy”.

Minor remarks

  1. I would like to suggest using the full name of the antibiotics seconeolitsine and  novobiocin instead of SCN and NOV, the number of the abbreviations used within the manuscript being large enough.

We have choose to maintain the abbreviations, mainly because they appear in several figures where the spacing is important.

  1. Lines 298 and 318 – species name should be in italics. OK, thank you. It has been changed

Round 2

Reviewer 3 Report

Comments and Suggestions for Authors

I am satisfied by the changes made by the authors so I accept the paper in its current form.